# Intimate partner violence screening during COVID-19

**Rebecka May Hoffman**[1]*, **Caitlin Ryus**[1], **Gunjan Tiyyagura**[2☯], **Karen Jubanyik**[1☯]

**1** Department of Emergency Medicine, Yale University School of Medicine, New Haven, Connecticut, United States of America, **2** Department of Pediatric Emergency Medicine, Yale University School of Medicine, New Haven, Connecticut, United States of America

☯ These authors contributed equally to this work.
* rebecka.hoffman@yale.edu

**Data Availability Statement:** The extent of the data poses a risk of re-identification of individuals and their associated HIPAA-protected data through deductive disclosure. The risk of re-identification precludes the data from being shared publicly. It

## Abstract

### Objectives

Emergency Department (ED) screening for intimate partner violence (IPV) is typically nursing-initiated, often with visitors present. Since the onset of the COVID-19 pandemic, we have seen both an increase in societal stress, a known exacerbator of IPV, and the implementation of visitor restriction policies. This combination presents the need for enhanced IPV screening and the opportunity to perform screening in a controlled, patient-only environment. Our goal was to evaluate the frequency of nurse-initiated screening for IPV prior to and during the early months of the COVID-19 pandemic as well as the frequency of positive screens for IPV.

### Methods

We conducted a retrospective cross-sectional study evaluating all adults (age >18 years) presenting to a tertiary care center ED. Patients were identified as presenting prior to the COVID-19 pandemic (June 1, 2019 to August 31, 2019) and after the COVID-19 visitor restriction policies (June 1, 2020 to August 31, 2020). Descriptive statistics were performed using chi-square and t-tests compared the demographic variables. Chi-square was used for a bivariate analysis of our primary outcomes (IPV screening performed and screening positive for IPV). Further analysis was done using a binary logistic regression model adjusting for the demographic characteristics.

### Results

Both the odds of nursing-initiated IPV screening and the odds of verbally screening positive for IPV significantly increased (OR 1.509, 95% CI 1.432–1.600) and (OR 1.375, 95% CI 1.126–1.681) respectively following the implementation of COVID-19 visitor restriction policies.

### Conclusions

These findings suggest that nurse-initiated IPV screening should continue to be performed with the patient privately, even after COVID-19 related ED visitor restrictions are removed.

may be made available upon request to users who agree to refrain from re-identifying individual study participants and who sign Data Use Agreement with Yale University. Those who are interested should discuss with the Yale Human Research Protection Program. Please contact Monika Lau, the Assistant Director for the Yale Human Research Protection Program, at monika.lau@yale.edu for further information.

**Funding:** This work was supported in part by funds from the National Institute of Child Health & Human Development grant K23HD107178 (GT). The contents of this manuscript are solely the responsibility of the authors and do not necessarily represent the official view of the NIH. The funders had no role in study design, data collection and analysis, decision to publish, or preparation of the manuscript.

**Competing interests:** The authors have declared that no competing interests exist.

These findings also support the hypothesis that the stress related to COVID-19 is contributing to a rise in IPV.

## Introduction

Intimate partner violence (IPV) is a serious and preventable public health issue that affects 1 in 3 women in the United States [1]. Victims of IPV suffer from higher rates of traumatic and atraumatic medical issues including depression, chronic pain, and post-traumatic stress disorder and overall have worse health outcomes [2, 3]. Screening for IPV is recommended by the US Preventive Services Task Force (USPSTF) for all women of child-bearing age [4]. Despite this, IPV screening rates in the ED are low [5, 6].

Societal stress historically exacerbates IPV and early reports during the COVID-19 pandemic suggest a similar increase in IPV [7]. Increased isolation from support networks, increased sequestration with a partner, and decreased access to services have been theorized as contributing factors to the rise in IPV [8–10]. Many studies have looked at this trend during the COVID-19 pandemic and commented on the importance of screening for IPV, however, to our knowledge, no study has looked at the rate of screening for IPV in the ED since COVID-19 [7–9]. With rising rates of IPV and the decrease in in-person medical care with the growth of tele-health (both prior to COVID-19 and exponentially so after COVID-19), finding opportunities for effective in-person IPV screening has become more important [7–9, 11].

In-person screening is most effectively done in private as partner interference in access to medical care is a known phenomenon in IPV [12]. COVID-19 provided an opportunity for this privacy as many EDs adopted new policies to mitigate infection spread such as strict visitor restriction rules at the onset of the pandemic. We theorized that the absence of visitors present due to the COVID-19 visitor restriction policies would improve IPV screening rates and that patients without visitors at the bedside would be more likely to self-report when screened, leading to a higher rate of positive IPV screens.

This study aimed to evaluate the both frequency of nurse-initiated screening for IPV at a single urban tertiary care center ED pre- and post-COVID-19 as well as the frequency of positive screens for IPV. Secondary aims included investigating whether demographic factors impact the rate of screening and the rate of positive screens.

## Methods

### Study design and setting

We performed a retrospective cross-sectional study using patient data from one ED at a large tertiary care center in an urban setting. We conducted study analyses according to the Strengthening the Reporting of Observational Studies in Epidemiology (STROBE) reporting guidelines for cross-sectional studies [13]. This study was approved by Yale University's Institutional Review Board. Informed consent was waived as the research posed no risk to the subjects.

### Selection of participants

Patients were included in this study if they were an adult (age >18) patient presenting to the ED during the specified study period. There were no exclusion criteria for the initial data collection. Patients were selected from the "pre-COVID-19" period, June 1, 2019- August 31,

2019, and the "COVID-19 visitor restriction" period, June 1, 2020- August 31, 2020. These dates were chosen after the completion of the first COVID-19 peak, a time when ED volumes and patient populations more closely reflected those pre-pandemic but the new visitor restriction policies were still in effect [14].

## Outcomes

Our primary independent variable was categorized as a binary variable which included the "pre-COVID-19" period and the "COVID-19 visitor restriction" period. Demographic variables of age, sex, gender, ethnicity, race, and insurance type were obtained from electronic medical record (EMR) data and were based on patient self-reporting at registration: sex, gender, ethnicity, race, and insurance type. (Table 1) Age was categorized into four categories: 18–33, 34–49, 50–64, and 65+. Sex was coded as male or female and gender identity was based on self-report at registration. A three-level variable was constructed from the sex/gender data fields: cis-male, cis-female, and gender minority (which included "Transgender Male/Female-to-Male", Transgender Female/Male-to-Female", "Genderqueer", "Intersex", "Gender non-conforming", "Other", and any patient whose sex was incongruous with their reported gender). Due to small sample size, gender minority was not disaggregated for analyses. Race/ethnicity categories were defined as White Non-Hispanic, Black Non-Hispanic, Latinx, Asian, First Nation, Multiracial, and unknown. Type of insurance was collapsed into four categories: private (BCBS, Commercial, Managed Care), public ("Medicare", "Medicaid", "Medicare Managed Care", "Medicaid Managed Care"), self-pay, or worker's compensation/other ("Worker's Comp", "Other", blank EMR entry).

**Table 1. Patient demographics.**

|  |  | Pre-COVID-19 | Post-COVID-19 | p-value |
|---|---|---|---|---|
| **Mean Age**** |  | **Mean (SD)** | **Mean (SD)** |  |
|  |  | 49.77 (19.87) | 49.95 (19.53) | 0.328 |
| **Sex %** |  | **n (%)** | **n (%)** |  |
|  | Male | 11,404 (48.7) | 9,895 (50.2) | 0.002* |
| **Self-Reported Gender** |  | **n (%)** | **n (%)** |  |
|  | Male | 11,375 (48.6) | 9,867 (50.0) |  |
|  | Female | 11,973 (51.1) | 9,787 (51.1) |  |
|  | Gender non-conforming | 79 (0.3) | 65 (0.3) |  |
| **Race/Ethnicity** |  | **n (%)** | **n (%)** |  |
|  | White or Caucasian | 11,626 (49.6) | 9509 (48.2) | 0.32 |
|  | Black or African American | 6,012 (25.7) | 5,256 (26.7) |  |
|  | Latinx | 4,788 (20.4) | 4,056 (20.6) |  |
|  | Asian | 428 (1.9) | 384 (1.9) |  |
|  | First Nation | 35 (0.1) | 19 (0.1) |  |
|  | Multi-racial | 395 (1.7) | 360 (1.8) |  |
|  | Unknown | 143 (0.6) | 135 (0.7) |  |
| **Insurance** |  | **n (%)** | **n (%)** |  |
|  | Private | 7,628 (32.6) | 6,320 (32.1) | 0.237 |
|  | Public | 14,267 (60.9) | 12,020 (61.0) |  |
|  | Self-pay | 1,005 (4.3) | 897 (4.5) |  |
|  | Worker's comp/other | 527 (2.2) | 482 (2.4) |  |

* denotes statistical significance at a p-value <0.05.

** denotes T-test, otherwise chi-square analysis was used.

The primary outcomes of interest were whether a patient was screened for IPV and whether they screened positive for IPV. The measure for IPV screening was a nurse-initiated screen that is a part of the standard ED triage assessment at our site. This IPV screen is comprised of a standardized series of verbal and non-verbal assessments (see Table 2 for screening questions) including three verbal screening questions that the nurse asks the patient and one non-verbal screening question for which the nurse assesses for objective signs of abuse on the patient (Table 2). Screening performed was determined if a response was entered by the nurse ("yes" or "no") for any of the four screening questions. Screening positive for IPV was defined as a "yes" answer to at least one of the screening questions.

Secondary outcomes of interest were related specifically to the screening items that required asking screening questions rather than relying on physical signs of abuse. For this secondary analysis, verbal screening was considered to have been performed if a response was entered by the nurse ("yes" or "no") for any of the verbal screening questions. Verbally screening positive for IPV was defined as a "yes" answer to at least one of those questions.

## Analysis

We performed descriptive statistics comparing demographic variables between the pre- and post-COVID-19 visitation policy cohorts using chi-square and t-tests. Initial bivariate analyses

**Table 2. Bivariate analysis of IPV screening and IPV screening positive pre- and post-COVID-19.**

| IPV Screening Rates Pre/Post-COVID-19 | | | Pre-COVID-19 | Post-COVID-19 | |
|---|---|---|---|---|---|
| | | | n (%) | n (%) | p-value |
| | | Screened | 19,963 (85.2) | 17,665 (89.6) | <0.001* |
| | | Not Screened | 3,464 (14.8) | 2,054 (10.4) | |
| **Screening Positive for IPV Pre/Post-COVID-19** | | | **Pre-COVID-19** | **Post-COVID-19** | |
| | | | n (%) | n (%) | p-value |
| | | Screened Positive△ | 225 (1.1) | 234 (1.3) | 0.082 |
| | | Screened Negative | 19,737 (98.9) | 17,431 (98.7) | |
| **Screening and Screening Positive Rates by Screening Questions** | | | | | |
| **Screening by Physical Signs** | | | **Pre-COVID-19** | **Post-COVID-19** | |
| | Physical signs of abuse present? | | n (%) | n (%) | p-value |
| | | Screened | 15,449 (65.9) | 15,146 (76.8) | <0.001* |
| | | Screening Positive | 57 (0.4) | 54 (0.4) | 0.924 |
| **Screening by Verbal Questions** | | | | | |
| | | Screened† | 18,139 (77.4) | 14,406 (73.1) | <0.001* |
| | | Screening Positive† | 187 (1.0) | 203 (1.4) | .002* |
| | Feels threatened by someone? | | | | |
| | | Screened | 17,897 (76.4) | 14,248 (72.3) | <0.001* |
| | | Screening Positive | 112 (0.6) | 128 (0.9) | 0.005* |
| | Feels unsafe at home/work/school? | | | | |
| | | Screened | 18,066 (77.1) | 14,436 (72.8) | <0.001* |
| | | Screening Positive | 140 (0.8) | 153 (1.1) | 0.006* |
| | Does anyone try to keep you from having contact with others or doing things outside your home? | | | | |
| | | Screened | 17,384 (74.2) | 14,138 (71.7) | <0.001* |
| | | Screening Positive | 26 (0.1) | 32 (0.2) | 0.145 |

* Denotes statistical significance at a p-value <0.05.

△ Screening positive is defined as a "yes" response to one or more of the nursing driven IPV screening questions.

† Composite of all verbal screening questions.

were performed comparing our primary outcomes (IPV screening performed and screening positive for IPV) between the pre- and post-COVID-19 visitation policy cohorts using chi-square. Binary logistic regression models adjusting for demographic characteristics were used to test the association between the visitation policy with our primary outcomes of IPV screening performed and screening positive for IPV. All analyses were performed using SPSS v28. All tests were 2-tailed and P < .05 was considered statistically significant.

## Results

### Primary outcomes

In the bivariate analysis using chi-square statistic, of the 23,427 patients identified in the 3-month pre-COVID-19 cohort, 85.2% were screened for IPV as compared to 89.6% of the 19,719 patients identified in the 3-month post-COVID-19 visitor policy cohort (p < .001). Screening positive for IPV did not change significantly post-COVID-19 visitor policy, (225 patients (1.1%) to 234 patients (1.3%) (p = .082)) (Table 2).

In a binary logistic regression model, screening had greater odds of taking place in the visitor restriction period than when visitors were allowed pre-COVID-19 (aOR 1.509, 95% CI 1.423–1.600) (Table 3). Once screened, patients had no significant change in the odds of screening positive for IPV in the visitor restriction period post COVID-19 than in the in the period prior to COVID-19 (aOR 1.179, 95% CI 0.980–1.418) (Table 3).

### Secondary outcomes by screening type

When examining verbal screening only, our team found that verbal screening decreased during the post-COVID visitor policy period from 77.4% of patients to 73.1% (p < .001). Of those who were verbally screened, the proportion of patients screening positive significantly increased between study periods from 1.0% to 1.4% (p = .001). Sub-group analysis of the specific screening question asked using chi-square showed a significant increase in positive screening rates for two questions: "Do you feel threatened by someone?" (0.6% to 0.9%, p = 0.005) and "Do you feel unsafe at home/work/school?" (0.8% to 1.1%, p = 0.006) (Table 2).

Binary logistic regression adjusting for demographic covariates demonstrated decreased odds of having been screened verbally in the visitor restriction period compared to prior (aOR 0.793, 95% CI 0.759–0.829). Among those verbally screened, there were greater odds for screening positive after the COVD-19 visitor policy compared to the pre-COVID period (aOR 1.375, 95% CI 1.126–1.681) (Table 3).

### Secondary outcomes by demographic variables

In the binary logistic regression model, when comparing to the youngest age cohort (age 18–33), all other age cohorts experienced significantly higher odds of being screened. All age cohorts also had significantly lower odds of screening positive for IPV than the 18–33 years cohort (Table 3).

Controlling for the other demographic covariates, identifying as cis-female or a gender minority was associated with significantly higher odds of the IPV screen being performed, aOR 1.142, 95% CI 1.07–1.209 and aOR 2.149, 95% CI 1.158–3.989 respectively. Cis-female patients also had a higher odds of screening positive for IPV (aOR 1.305, 95% CI 1.081–1.575). There was no significant difference in the odds of having a positive screen for gender-minority patients (Table 3).

As compared to White Non-Hispanic patients, many racial and ethnic minorities were screened less for IPV including Black Non-Hispanic (aOR 0.775, 95% CI 0.721–0.832), Latinx

**Table 3. Fully adjusted binary logistic regression analysis of (1) Verbal IPV screening pre- and post-COVID-19, (2) Verbal IPV screening positive vs negative, (3) IPV screening pre- and post-COVID-19, (4) IPV screening positive vs. negative.**

| Odds Ratios (95% Confidence Intervals) | | | Verbal IPV Screen Performed | Verbal IPV Screened Positive for IPV | IPV Screen Performed | Screened Positive for IPV |
|---|---|---|---|---|---|---|
| | | | n = 32,545 | n = 390 | n = 37,628 | n = 459 |
| **COVID-19 visitor restriction** | | | aOR (CI) | aOR (CI) | aOR (CI) | aOR (CI) |
| | | | 0.793 (.759–0.829)* | 1.375 (1.126–1.681)* | 1.509 (1.423–1.600)* | 1.179 (0.980–1.418) |
| **Age** | | | aOR (CI) | aOR (CI) | aOR (CI) | aOR (CI) |
| | 18–33 years | | 1 [Reference] | 1 [Reference] | 1 [Reference] | 1 [Reference] |
| | 34–49 years | | 1.098 (1.033–1.167)* | 0.793 (0.614–1.023) | 1.142 (1.060–1.231)* | 0.780 (0.619–0.984)* |
| | 50–64 years | | 1.265 (1.188–1.347)* | 0.625 (0.475–0.822)* | 1.472 (1.358–1.596)* | 0.562 (0.435–0.726)* |
| | 65+ years | | 1.045 (0.980–1.114) | 0.336 (0.243–0.465)* | 1.515 (1.392–1.650)* | 0.290 (0.214–.393)* |
| **Self-Reported Gender** | | | aOR (CI) | aOR (CI) | aOR (CI) | aOR (CI) |
| | Male | | 1 [Reference] | 1 [Reference] | 1 [Reference] | 1 [Reference] |
| | Female | | 1.096 (1.049–1.146)* | 1.486 (1.209–1.827)* | 1.142 (1.07–1.209)* | 1.305 (1.081–1.575)* |
| | Gender Minority | | 1.758 (1.131–2.732)* | 1.432 (0.349–5.883) | 2.149 (1.158–3.989)* | 1.141 (0.279–4.668) |
| **Race/Ethnicity** | | | aOR (CI) | aOR (CI) | aOR (CI) | aOR (CI) |
| | White Non-Hispanic | | 1 [Reference] | 1 [Reference] | 1 [Reference] | 1 [Reference] |
| | Black Non-Hispanic | | 0.853 (0.807–0.901)* | 0.824 (0.648–1.048) | 0.775 (0.721–0.832)* | 0.795 (0.636–0.993)* |
| | Latinx | | 0.907 (0.853–0.965)* | 0.538 (0.400–0.724)* | 0.833 (0.770–0.901)* | 0.543 (0.414–0.711)* |
| | Asian | | 0.866 (0.737–1.018) | 0.490 (0.181–1.328) | 0.908 (0.733–1.124) | 0.399 (0.148–1.079) |
| | First Nation | | 1.274 (0.640–2.538) | NA† | 1.332 (0.528–3.361) | NA† |
| | Multiracial | | 0.780 (0.663–0.918)* | 1.030 (0.522–2.031) | 0.749 (0.611–0.919)* | 0.910 (0.478–1.731) |
| | Unknown | | 0.849 (0.649–1.111) | NA† | 0.647 (0.469–0.893)* | 0.329 (0.046–2.362) |
| **Insurance** | | | aOR (CI) | aOR (CI) | aOR (CI) | aOR (CI) |
| | Private | | 1 [Reference] | 1 [Reference] | 1 [Reference] | 1 [Reference] |
| | Public | | 0.912 (0.868–0.959)* | 2.610 (1.999–3.409)* | 0.874 (0.818–0.934)* | 2.559 (2.002–3.273)* |
| | Self-Pay | | 0.831 (0.743–0.929)* | 1.446 (0.755–2.767) | 0.730 (0.637–0.836)* | 1.947 (1.153–3.288)* |
| | Other | | 0.754 (0.654–0.869)* | 2.696 (1.451–5.008)* | 0.653 (0.550–0.777)* | 2.373 (1.314–4.284)* |

* Denotes statistical significance.

† Unable to calculate due to lack of patients screening positive.

(aOR 0.833, 95% CI 0.770–0.901), and Multi-racial (aOR 0.749, 95% CI 0.611–0.919) (Table 3).

Patients with public, self-pay, or other forms of insurance had lower odds of being screened for IPV. Once screened, patients with public insurance were almost 150% more likely to screen positive for IPV as compared to those with private insurance (aOR 2.479, 95% CI 1.693–3.629) in the binary logistic regression model (Table 3).

## Discussion

Our study was the first to investigate the effect of the COVID-19 visitor restriction policies on IPV screening in the ED. We demonstrated that following the implementation of the visitor restriction policies, the odds of nursing-initiated IPV screening occurring significantly increased and the odds of verbally screening positive for IPV significantly increased.

Screening for IPV is a primary modality for IPV case identification in the healthcare setting. The ED provides healthcare professionals a unique opportunity to screen and intervene on IPV. ED patients are seen regardless of demographic factors, and visits occur in person. Our hospital has a nurse-initiated IPV screen that is a part of the initial triage assessment. This

screen is often completed in a way that does not facilitate IPV disclosure; visitors, who are potentially the perpetrator of the IPV, are typically allowed to stay at bedside during the questioning [15]. Since the onset of the COVID-19 pandemic, increases in the occurrence of IPV as well as the implementation of visitor restriction policies, provide both the necessity and the opportunity for improved IPV screening [7–9].

Many studies have shown that IPV has increased during COVID-19, however, to our knowledge no studies have looked specifically at IPV screening during this time period [7–9]. During the unique time of COVID-19, when patients rarely had visitors at bedside due to visitor restrictions, we demonstrated an increase in screening for IPV. This novel finding supports prior qualitative studies that show a lack of privacy as a barrier to IPV screening [16]. The increase in overall screening was largely due to an increase in screening for physical signs of abuse as an adequate physical exam requires privacy for completion. Of note, verbal screening decreased during our studied time period and this is possibly the result of changes in triaging priorities and provider-patient interactions during the COVID-19 pandemic.

Positive IPV disclosures in response to verbal screening also increased post-COVID-19 visitor restriction. We hypothesize that this is the result of improved screening secondary to privacy and increased rates of IPV due to pandemic associated stress [7, 9]. Adequate privacy may both reduce the stigma associated with disclosing IPV and the influence that the visitor may have on the screening, especially if they are the abuser [12, 17]. Lack of privacy has also been cited as a barrier to the provider initiating the screen [16].

In the ED, patients often have visitors at bedside when the nurse-initiated IPV screen is conducted, an environment that is not conducive for disclosure [15]. Studies have shown that privately screening in EDs through means such as technology increases both screening and disclosures for IPV [18]. In the future, once COVID-19 visitor restrictions are lifted, IPV screening should continue to occur with adequate privacy.

While it has been suggested that racial and ethnic minorities are at a higher risk for experiencing IPV, various studies reporting results of multi-variate analyses investigating this relationship have provided conflicting findings [1, 19–23]. In our study, Black Non-Hispanic, Latinx, and Multiracial patients were significantly less likely than White Non-Hispanic patients to be screened for IPV. Black Non-Hispanic and Latinx patients were also significantly less likely than White Non-Hispanic patients to screen positive for IPV. Possible reasons for these findings include differences in access to medical care, perceptions of responses to disclosures by healthcare providers, cultural differences, and language barriers [16, 24]. It has previously been shown that non-English preferring patients are less likely to be screened for IPV than English preferring patients [20]. Further analysis to understand the interaction of primary language preference and race and ethnicity as well as the impact of limited English proficiency and IPV disclosure would be beneficial to inform more culturally sensitive screening practices in EDs.

Many studies have shown that IPV disproportionately impacts those that are socioeconomically disadvantaged, and that this disparity has been augmented by the COVID-19 pandemic [19, 21, 22, 25, 26]. Studies are mixed on the relationship between socio-economic status and the frequency of IPV screening [20, 21]. We demonstrated that non-privately insured patients were significantly less likely to be screened for IPV while being significantly more likely to screen positive for IPV. This highlights a critical need to improve IPV screening in this disproportionally affected population. Providers should take note of potential underlying biases that could lead to decreased screening in this high-risk population. Further exploration into discrepancies of screening is necessary.

### Limitations

There were at least four main limitations to this study. First, our findings suggest that screening patients in isolation increases both the rate of nurse-initiated IPV screening as well as the yield of verbal IPV screening; however, given the cross-sectional study design, we are unable to draw causal inference. Second, our study only examined the nurse-initiated IPV screening questions and was not designed to detect IPV identification by other health care providers in the ED. Third, we assumed that all patients in the pre-COVID-19 period had visitors and all patients in the treatment group were screened in isolation. Finally, this is a single center study, however, the patterns demonstrated by the demographic analysis are consistent with previous findings on screening and disclosure rates for IPV.

## Conclusions

In summary, this study underscores the importance of screening ED patients for IPV in privacy. It also suggests the need to educate health care providers on the importance of screening populations that have been shown to suffer disproportionately from IPV. Societal stress has historically led to increases in IPV and our data supports a limited but rapidly growing body of evidence showing concerning trends during the COVID-19 pandemic [7–9]. With increases in the use of tele-health, the ED may be one of the few places IPV victims receive in-person care [11]. Screening and providing support for IPV should remain a priority for Emergency providers.

## Author Contributions

**Conceptualization:** Rebecka May Hoffman, Gunjan Tiyyagura, Karen Jubanyik.

**Data curation:** Caitlin Ryus.

**Investigation:** Rebecka May Hoffman.

**Methodology:** Rebecka May Hoffman, Caitlin Ryus, Gunjan Tiyyagura, Karen Jubanyik.

**Supervision:** Rebecka May Hoffman, Gunjan Tiyyagura, Karen Jubanyik.

**Writing – original draft:** Rebecka May Hoffman.

**Writing – review & editing:** Rebecka May Hoffman, Caitlin Ryus, Gunjan Tiyyagura, Karen Jubanyik.

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
