## [Decision Letter · Decision Letter 0]

23 Jan 2023

PONE-D-22-27460Intimate partner violence screening and rates of screening positive post-COVID-19 visitor restrictionPLOS ONE

Dear Dr. Hoffman,

Thank you for submitting your manuscript to PLOS ONE. After careful consideration, we feel that it has merit but does not fully meet PLOS ONE’s publication criteria as it currently stands. Therefore, we invite you to submit a revised version of the manuscript that addresses the points raised during the review process. Firstly, I would suggest you to kindly go through the standard author guidelines of the journal. Kindly follow the suggestions related to both abstract and manuscript. The abstract does not clearly mention the methodology guidelines especially statistical analysis. Furthermore, in the main manuscript, it mentions that submission is made based on STROBE checklist however, there are various inadequacies. Firstly, the introduction needs more expansion. A strong rationale/justification of the study needs to be highlighted. In the methods section, clearly mention the details of bivariate and multivariate analysis, sample size, sampling etc. The limitations should be a mentioned at the end of discussion. It is advisable to go through ICJME guidelines of manuscript submission. The discussion seems inadequate and the major study findings need to be discussed more (both before and during COVID changes). 

We look forward to receiving your revised manuscript.

Kind regards,

Reshu Agrawal Sagtani

Academic Editor

PLOS ONE

Journal Requirements:

Reviewers' comments:

Reviewer's Responses to Questions

**Comments to the Author**

1. Is the manuscript technically sound, and do the data support the conclusions?

Reviewer #1: Yes

Reviewer #2: Partly

2. Has the statistical analysis been performed appropriately and rigorously? 

Reviewer #1: Yes

Reviewer #2: I Don't Know

3. Have the authors made all data underlying the findings in their manuscript fully available?

Reviewer #1: Yes

Reviewer #2: Yes

4. Is the manuscript presented in an intelligible fashion and written in standard English?

Reviewer #1: Yes

Reviewer #2: No

5. Review Comments to the Author

Reviewer #1: Title: The study title needs to be revised.

Study Participants: How do you ascertain that participants are in relationship?

Outcome: As it is mentioned that it is a single center study, were there different study sites?

Analysis(We performed descriptive statistics comparing demographic variables between the pre- and post 106 COVID-19 visitation policy cohorts using chi-square and t-tests.):

:Were your analysis limited to descriptive statistics?

Discussion (Despite this, and consistent with 194 previous studies, we demonstrated that Latinx, Black Non-Hispanic, and Multiracial patients 195 compared to White Non-Hispanic patients were significantly less likely to be screened for IPV)

: Can you give any reason for the same?

References: Please consider adding few more references.

Reviewer #2: The paper needs a thorough revision for language, clarity, grammar, etc. Request to write the manuscript in the simple and clear language.

Introduction

The introduction needs significant expansion, focusing specifically on what gaps in the literature this study is trying to address? Why this study is important?

Results

The result section is quite unclear and need significant explanation in terms of the use of chi sqare test and t test.

Discussion

The discussion is lacking as it relates to unpacking what your results mean. How is this novel or important? What are the main takeaways for the reader? This section needs to be better developed.

6. PLOS authors have the option to publish the peer review history of their article (what does this mean?). If published, this will include your full peer review and any attached files.

Reviewer #1: No

Reviewer #2: No

---

## [Author Response · Author response to Decision Letter 0]

27 Feb 2023

Academic Editor

1. The abstract does not clearly mention the methodology guidelines especially statistical analysis.

a. Thank you, we have added to the abstract to provide clarity on our statistical analysis

2. in the main manuscript, it mentions that submission is made based on STROBE checklist however, there are various inadequacies

a. Thank you, we feel that by responding to the valuable feedback by the reviewers our paper meets STROBE guidelines

3. Firstly, the introduction needs more expansion. A strong rationale/justification of the study needs to be highlighted.

Thank you for this comment! We hope that our expansion on our introduction has provided more clarity to the topic and highlighted the rationale for the study.

4. In the methods section, clearly mention the details of bivariate and multivariate analysis, sample size, sampling etc. 

Thank you. Based on your feedback we have expanded upon our methods section and feel it has greatly strengthened our paper. In addition, given no prior literature evaluating IPV screening during COVID-19, we did not have an a priori estimate for a difference in means between screening before and after the COVID-19 period. However, based on sample size calculations, a population of approximately 7,000 was sufficient to demonstrate a 1% difference in means with 80% power and a level of significance of 5%.

5. The limitations should be a mentioned at the end of discussion.

Thank you! We have moved this section

6. The discussion seems inadequate and the major study findings need to be discussed more (both before and during COVID changes)

We appreciate this feedback and we feel our expanded discussion has improved out manuscript.

Thank you for the links to the template! These were very helpful in editing our manuscript to meet PLOS ONE’s style requirements.

8. In your Data Availability statement, you have not specified where the minimal data set underlying the results described in your manuscript can be found.

We have included a letter from Deputy HIPAA Privacy Officer for Research in regards to this topic. 

9. Please include your full ethics statement in the ‘Methods’ section of your manuscript file. In your statement, please include the full name of the IRB or ethics committee who approved or waived your study, as well as whether or not you obtained informed written or verbal consent. If consent was waived for your study, please include this information in your statement as well. 

Yes, we agree on the importance of including this. Please see line 105 “Yale University’s Institutional Review Board. Informed consent was waived as the research poses no risk to the subjects.”

Reviewer 1

10. The study title needs to be revised.

Thank you, we have updated our title to “Intimate partner violence screening during COVID-19”

11. Study Participants: How do you ascertain that participants are in relationship?

Thank you for your comment! We were unable to ascertain if participants were in a relationship. All patients presenting to our ED are screened as a part of the nursing triage assessment regardless of relationship status. Please see the methods section, line 110.

12. Outcome: As it is mentioned that it is a single center study, were there different study sites?

Thank you for your question! This study was completed at a single urban tertiary care center ED. We have added additional language that we hope adds clarity. Please see the methods section, line 102.

13. Analysis(We performed descriptive statistics comparing demographic variables between the pre- and post 106 COVID-19 visitation policy cohorts using chi-square and t-tests.):

:Were your analysis limited to descriptive statistics?

Thank you for your comment! No, in addition to the descriptive statistics, we performed a binary logistic regression model. We have added additional language to the manuscript for clarity.

14. Discussion (Despite this, and consistent with 194 previous studies, we demonstrated that Latinx, Black Non-Hispanic, and Multiracial patients 195 compared to White Non-Hispanic patients were significantly less likely to be screened for IPV)

: Can you give any reason for the same?

Thank you, based on your feedback we have significantly expanded upon this paragraph and we think our manuscript is stronger for it. Please see the paragraph beginning at line 357.

15. References: Please consider adding few more references.

Thank you! With the additions to the manuscript based on the other comments, we added several more references

Reviewer 2

16. The paper needs a thorough revision for language, clarity, grammar, etc. Request to write the manuscript in the simple and clear language.

Thank you for your feedback! We feel that our further edits for language and grammar in addition in addition to responding to helpful feedback here has greatly strengthened our paper.

17. Introduction

The introduction needs significant expansion, focusing specifically on what gaps in the literature this study is trying to address? Why this study is important?

Thank you for this comment. We have greatly expanded the introduction, and focused on the gaps in the literature that our study is attempting to address. For example, please see line 74: “Increased isolation from support networks, increased sequestration with a partner, and decreased access to services have been theorized as contributing factors to the rise in IPV. Many studies have looked at this trend during the COVID-19 pandemic and commented on the importance of screening for IPV, however, to our knowledge, no study has looked at the rate of screening for IPV in the ED since COVID-19.”

18. Results

The result section is quite unclear and need significant explanation in terms of the use of chi square test and t test.

Thank you for your comment! We hope that the additions made to the results section as well as table 1 have added clarity.

19. Discussion

The discussion is lacking as it relates to unpacking what your results mean. How is this novel or important? What are the main takeaways for the reader? This section needs to be better developed

Thank you for your valuable feedback. We hope the additions we made to our discussion section drive our message home. For example, please seen line 357: “Many studies have shown that IPV has increased during COVID-19, however, to our knowledge no studies have looked specifically at IPV screening during this time period.[7–9] During the unique time of COVID-19, when patients rarely had visitors at bedside due to visitor restrictions, we demonstrated an increase in screening for IPV. This novel finding supports prior qualitative studies that show a lack of privacy as a barrier to IPV screening”

---

## [Editor Report · Decision Letter 1]

27 Mar 2023

Intimate partner violence screening during COVID-19

PONE-D-22-27460R1

Dear Dr. Hoffman,

We’re pleased to inform you that your manuscript has been judged scientifically suitable for publication and will be formally accepted for publication once it meets all outstanding technical requirements.

Kind regards,

Reshu Agrawal Sagtani

Academic Editor

PLOS ONE
---

## [Editor Report · Acceptance letter]

14 Apr 2023

PONE-D-22-27460R1 

Intimate partner violence screening during COVID-19 

Dear Dr. Hoffman:

I'm pleased to inform you that your manuscript has been deemed suitable for publication in PLOS ONE. Congratulations! Your manuscript is now with our production department. 

Kind regards, 

on behalf of

Dr. Reshu Agrawal Sagtani 

Academic Editor

PLOS ONE